# A Modeling Framework for the Thermoforming of Carbon Fiber Reinforced Thermoplastic Composites

**DOI:** 10.3390/polym16152186

**Published:** 2024-07-31

**Authors:** Long Bin Tan, Wern Sze Teo, Yi Wen Cheah, Sridhar Narayanaswamy

**Affiliations:** 1Institute of High Performance Computing (IHPC), Agency for Science, Technology and Research (A*STAR), 1 Fusionopolis Way, #16-16, Connexis North Tower, Singapore 138632, Singapore; narayanas@ihpc.a-star.edu.sg; 2Singapore Institute of Manufacturing Technology (SIMTech), Agency for Science, Technology and Research (A*STAR), 73 Nanyang Drive, Singapore 637662, Singapore; wsteo@simtech.a-star.edu.sg (W.S.T.); cheah_yi_wen@simtech.a-star.edu.sg (Y.W.C.)

**Keywords:** carbon fiber reinforced polymers (CFRP), thermoforming, defects, wrinkling, thermoplastic composites, numerical simulations

## Abstract

A comprehensive modeling framework for the thermoforming of polymer matrix woven laminate composite was developed. Two numerical indicators, the slip path length and traction magnitude, have been identified to be positively correlated to matrix smearing and wrinkling defects. The material model has been calibrated with picture-frame experimental results, and the prediction accuracy for intra-ply shear and thickness distribution was examined with measurements of the physically formed parts. Specifically, thickness prediction for most locations on the formed parts was accurate within an 11.6% error margin. However, at two points with significant intra-ply shear, the prediction errors increased to around 20%. Finally, a parametric study was conducted to determine the relationship between various process parameters and the quality of the formed part. For the trapezoidal part, orienting the laminate at 45 degrees to the mold axis reduces the likelihood of matrix smear and wrinkling defects. Although this laminate orientation yielded a greater spatial variation in part thickness, the thickness deviation is lower than that for the 0-degree orientation case. Two forming analyses were conducted with ramp rates of 25 mm/s and 80 mm/s to match the equipment’s operational limits. It was observed that higher forming rates led to a greater likelihood of defects, as evidenced by a 15% and 10% increase in the formed part areas with longer slip paths and higher traction magnitudes, respectively. It was discovered that shallower molds benefit from faster ramp rates, while deeper molds require slower rates to manage extensive shearing, stretching and bending. Faster forming rates lead to smaller thickness increases at high intra-ply shear regions, indicating a shift from intra-ply shear to out-of-plane bending due to the visco-plastic effect of the molten laminate and can negatively impact part quality. Lastly, it was shown that a well-conceived strategy using darts could improve the part quality by reducing the magnitude of the defect indicators.

## 1. Introduction

In recent years, there has been a resurgence of interest in thermoplastic composites, driven by the growing emphasis on sustainability and material recyclability. The unique properties of thermoplastics, such as their ability to undergo multiple melt and solidification cycles without significant degradation, offer new avenues for production process design. This includes opportunities for joining or reshaping semi-finished parts, as well as the ease of repairing localized cracks and delamination [1]. The absence of curing time in thermoforming pre-consolidated laminates also streamlines the production cycle, making it well-suited for automated mass production [2].

For fiber-reinforced composites, structural optimization is facilitated by the ability to orient fibers in their optimal directions for strength and performance, while rapid cooling rates enable tailored polymer crystallization. Industries such as aerospace and automotive are actively exploring fast and cost-efficient processing technologies, particularly in press forming, tape placement, winding and joining processes [2].

Several thermoplastic resins, such as the semi-crystalline PA12, are gaining popularity in both the oil and gas and the automotive industries, while PEEK is well-known in high-end applications in the aerospace and medical industry due to its excellent mechanical strength and resistance to high temperature and hydrolysis [2]. Excellent laminate properties of PPS, PEKK and PEEK, together with carbon or glass fiber reinforcement, have been reported [3], and thermoplastic composite stiffeners were also reportedly used in the wing and fuselage assemblies of aircraft [4] due to their toughness and suitability for crash applications [5]. Carbon fiber reinforced polymer (CFRP) composite bars have also been reported to exhibit outstanding corrosion resistance, fatigue performance and high specific strength and modulus. They have also gained interest as a substitute for steel reinforcement bars in concrete structures to combat corrosion and improve durability. Recent research by Xian et al. [6] investigated the bending properties of similar composite bars, which is essential for their engineering application. Vedernikov et al. [7] found that in-house pultruded pre-consolidated tape (PCT) flat laminates yielded twice the flexural strength and about 28% higher interlaminar shear strength compared to commercially produced laminates, likely due to better impregnation of the fiber reinforcements for the in-house PCTs.

This paper presents a comprehensive modeling framework developed for the thermoforming process of woven carbon fiber thermoplastic composite parts. It outlines a range of numerical analyses designed to streamline setup procedures, ultimately leading to reduced design cycle times and decreased reliance on physical trial and error. Furthermore, a detailed parametric study is undertaken to provide valuable insights into the optimization of process parameters, with the ultimate goal of enhancing part quality.

## 2. Literature Review on Thermoforming Carbon Fiber Reinforced Plastics

Thermoforming plays a crucial role in various manufacturing strategies, whether it involves creating components for larger assemblies or serving as protective covers for products. Being an inclusive and versatile process that can relate to many industries, a deep understanding of the physics behind laminate forming is essential for achieving defect-free parts [3,8]. 

Early fabrication of fiber composite parts relied on manual lay-up of fabric mats followed by resin application. While this method could mitigate wrinkle formation and surface defects to some extent and steer the fibers in favorable directions [8], it was labor-intensive and unsuitable for mass production. As a result, rapid manufacturing processes like hot press forming of thermoplastic composites have emerged as key technologies for achieving high-volume, large-format and consistent quality production. 

Currently, the thermoforming of carbon fiber reinforced polymers (CFRP) often involves extensive trial and error methods to minimize defects such as part thinning, surface smears, wrinkles and shape distortions [9]. This process is notably more complex compared to working with pure polymers, primarily due to the woven fabric nature of the composite materials and the limited understanding of forming technologies for prepreg materials, along with the absence of forming parameter specifications from materials manufacturers [9]. Moreover, the manufacturing of tools is time-consuming and costly, significantly impacting the development costs of formed parts. To ensure the success of products in the market, it is imperative to mitigate the need for trial-and-error development approaches. 

Simulations of composite forming processes can potentially reduce costly iterations, help to evaluate alternative process steps, and are crucial in forming a mature position of composites in this market for high-volume manufacturing [8]. By employing formability analyses, it becomes possible to anticipate and address production issues early in the design phase, thus minimizing the need for expensive tool modifications during testing. The analyses are invaluable in predicting various aspects of the formed part, including geometry, fiber directions, thickness distribution, shape distortions and defects, and can also support the development of design guidelines. In the early stages of the design phase, these guidelines are particularly beneficial, as they aid in defining materials and initial process parameters [9,10,11]. A process window or a ranking of materials based on their formability can serve as a useful tool [3], yet such ranking or guidelines are currently lacking for thermoplastic composites. Recently, Loh et al. [12] performed close to 1500 thermoforming simulations on the double-dome and trapezoidal profile and derived process windows and design criteria for the plain-woven CFRTP laminates. The development of software and theoretical models relies heavily on experimental data for validation and calibration, which is not always comprehensively reported [3,8]. This dependency underscores the importance of having accepted material characterization methods and associated material property databases to accommodate the growing range of materials available on the market. The models reported thus far are primarily isothermal, but non-isothermal models are becoming increasingly essential to accurately predict shape distortions resulting from process-induced residual stresses and temperature-related defects, such as warpage due to mold and part cooling. Hence, the current predictive capabilities and limitations of such simulation tools remain uncertain, with only a limited number of studies reporting both theoretical work and detailed experimental validation.

Sebastiaan [3] conducted a sensitivity study to examine the impact of various material parameter combinations on the formability of a quasi-isotropic laminate onto a doubly curved dome geometry. Different values of inter-ply friction, as well as ply bending and shear stiffnesses, were investigated, which concluded that the out-of-plane wrinkling of a formed part is directly related to the relatively lower bending stiffness of the ply over its shear stiffness such that during the pressing phase, the ply is more susceptible to wrinkling [4]. In a related study, Boisse et al. focused on the response of fabric under biaxial tension along fiber directions [11], measuring the experimental nonlinear response and incorporating it into a forming simulation model. Subsequently, the focus shifted towards studying the in-plane trellis shear response, with results being normalized for different specimen sizes. Modeling efforts were directed towards understanding the rate-independent behavior of dry fabrics characterized by normalized shear force as a function of shear angle [13]. Exploring the coupled effects of these factors can be facilitated through the application of Machine Learning (ML) or Artificial Intelligence (AI) techniques where Tan et al. [14,15] presented such approaches in inverse prediction of the process parameters that formed a final part, and also in obtaining the multi-parameter optimization for composite thermoforming. While simulations have been increasingly adopted by industries to optimize the thermoforming process and ensure part quality, their use does not guarantee infallibility, and further research is needed to comprehensively define the physics and interactions that may be occurring in such processes.

## 3. Motivation

There is currently no literature that comprehensively details the framework for thermoforming of CFRTP composites, the various simulations that can be conducted to support different phases of the thermoforming process, and the extensive correlation of measurement data with numerical analysis. This paper presents a consolidated effort to provide experimental data to explore parameter relationships and sensitivities through parametric study and in devising ways, such as through darts, to minimize part defects. 

## 4. Part Geometry Selection and Thermoforming Process

The trapezoidal and cylindrical dome profiles were selected for use in our study. Their corresponding male and female molds were manufactured for the thermoforming experiments. Figure 1 shows the setup in the thermoforming simulation, which reflects the actual thermoforming process conducted in SIMTech (See Figure 2). The mold plates have corner curvatures of 2 mm and are 2D rigid geometries that press the laminate into shape. The trapezoidal molds have inclined front and side surfaces, forming draft angles of 30 and 10 degrees from the vertical, respectively, with a horizontal top surface. The male cylindrical dome has a dome cap and cylindrical surface that is 30 mm in radius. The female mold geometries are created by offsetting the male geometry outwards by the thickness of the laminate. The length of the parts, excluding the flanges, is 130 mm. 

The laminate is gripped and suspended by four corner spring tensioners so as to transport it between the heating stage and the forming stage. The spring stiffness of the tensioners is measured to be about 0.126 N/mm. The forming aspect ratio, i.e., the shortest width of the part versus the formed height of the part, is 1:1. The ratio is very high, as most thermoforming parts usually have an aspect ratio of 2:1 or below. The high ratio allows the forming defects to occur more readily so as to study and find ways to mitigate them. In the analysis, the transport frame and the molds are assumed to be rigid as they are made of stainless steel and have much higher stiffness than the heated molten laminate. The laminate comprises four plies of twill-woven carbon-fiber reinforced thermoplastic (CFRTP) that are each 0.3125 mm thick. The total initial thickness of the laminate is 1.25 mm.

The thermoforming analysis comprises three steps. The first step has an analysis time of four (4) seconds, with only gravity acting on the laminate suspended by the spring tensioners and the transport frame. This simulates the initial sagging of the molten laminate after being heated to its forming temperature at the heating stage and the time required to transport the laminate to the forming stage. The second step corresponds to the lowering of the top female mold, at 120 mm/min, to almost touch the laminate, while the third step corresponds to the press-forming of the laminate, at 25 mm/min, into the desired profile. Figure 2a–c show the physical forming process as performed in SIMTech. Figure 2d shows the formed part when the mold opens. At this stage, any defects such as wrinkling or surface blemish could be checked. Figure 2e,f show examples of formed trapezoidal and cylindrical dome parts with visible defects. 

## 5. Materials and Methods

The thermoforming analysis was performed using the state-of-the-art finite element-based thermoforming software AniForm^TM^ (Version 4.2). The rigid mold CADs were imported into the software before the material definitions, constraints and load conditions were applied. The thermoforming behavior of the laminate is mathematically captured using various material models. Essentially, the mechanisms observed in composite laminate forming are intra-ply shear and bending, intra-ply extension and compression in the fiber and transverse directions, and tool–ply and ply–ply interfacial slippage [3]. Table 1 shows the properties used for the thermoforming analysis. The general properties are Poisson ratio, υ, and density, ρ. The ply tensile, in-plane shear and out-of-plane bending moduli are given as E_F_, E_TI_ and E_TB_, respectively, while η_0_ and η_∞_ are predefined zero shear rate and infinite shear rate viscosities. 

The isotropic elastic and the Cross-Viscosity models are jointly used to predict the in-plane and bending behaviors of the plies. The Cross model is a shear-rate-dependent viscosity model, as found in the work of Macosko [16]. It is used to model the power law type of response with a viscosity plateau region at low and high shear rates, respectively. The mathematical equations below describe the Cross model: $$\eta \left(\dot{\gamma }\right)=\frac{\eta_{0}-\eta_{\infty }}{1+m{\dot{\gamma }}^{1-n}}+\eta_{\infty }$$
$$\mathrm{And}\;\sigma =\frac{2\eta (\dot{\gamma )}}{J}$$
where **σ** is the Cauchy stress, **D** is the rate of deformation tensor, and J is the Jacobian of the deformation gradient.

A mixed model that combines the two models in parallel is used to describe the overall deformation mechanism. The total stress response is equal to $\sigma =\sum_{i=1}^{n}\upsilon_{i}\sigma i$, where each basic model stress tensor, ***σ****_i_*, can be scaled by weight. In our modeling, we assume equal weightage between the elastic and viscous response, i.e., υ_1_ = υ_2_ = 1. The material parameters for ply in-plane behavior are obtained from curve fitting of the picture-frame test data. Procedures for the ply tests are available in the literature [17]. After the material parameters are extracted, the picture-frame analysis is set up in AniForm^TM^ to re-simulate the test process and verify with the load–displacement curve, ply shear and thickness of the samples as an additional model check. Figure 3 shows the validation of the model’s mechanical response with test data, while Figure 4 shows the optical micrographs at five locations (two near the top, two near the bottom and one at the center) of the tested samples, and Table 2 shows the tabulated thickness and shear angle measurements after test. It is noted that the shear angle obtained from simulation at the test region was 52.6°, which is very close to those obtained from test that ranges from 48° to 53°. However, the measured thicknesses are around 0.63 mm, which is 14.5% higher than the thickness of 0.55 mm, as predicted from the simulation. A bulk equivalent model, as stated in the preceding paragraph, was used to model the ply response. But realistically, the increase in thickness is caused by the change in weave angles of the initially perpendicular fibers. The ply is not of uniform thickness but is thicker in the overlapped regions and thinner in other regions. The original ply thickness was 0.3125 mm, and it is noteworthy that the applied intra-ply shear increases this thickness by nearly 2 times. 

The penalty–polymer model, which includes penalty contact in combination with a viscous type of friction, is used to model the interfacial contact between the plies and between the outer plies and the mold interior surfaces. Standard interfacial material parameters, as supplied by AniForm^TM^, were used in the analyses due to the lack of interfacial test data at the time of writing. However, sensitivity analysis was conducted with the thermoforming simulation, which showed that the results of the part formability do not change significantly with these parameters. 

## 6. Thermoforming Mechanism and Defect Indicators

Figure 5 illustrates the progressive deformation process of the laminate during thermoforming. Initially, the heated CFRTP laminate is transported to the forming stage, where it begins to sag slightly into a U-shaped pattern along the x-axis due to its weight and the temporary loss of stiffness caused by elevated temperature. The shape of the sagging depends on factors such as the layout of spring tensioners and the pretension used.

As the laminate sag stabilizes, the top female mold moves downward to touch the top surface of the laminate (see Figure 5a). Further pressing of the female mold causes the laminate region within the mold to flex upwards against gravity. Initial contact between the laminate and the mold occurs at the bends as the mold drapes the laminate over (see Figure 5b). The primary traction and slippage occur at the region where the laminate folds within the gap space of the closing female mold with the male mold.

While contact at the internal bend (bottom surface of the laminate with edges of the male mold) remains relatively unchanged, contact at the external bend (top surface of the laminate with edges of the female mold) is constantly shifting as the laminate stretches and drapes over the external surface of the male mold (see Figure 5c). This continuous movement can potentially lead to surface defects on the top surface of the laminate due to prolonged contact with the cooler female mold surface. As the forming nears completion, the bulge at the laminate center is pressed to the required thickness while pushing material further out to the side walls (see Figure 5d), which at this time will experience greater frictional contact. The press process continues until a full load is applied, and where the laminate at the mold flanges is also completely pressed.

Figure 6 (top) shows that near the sloped wall region of the trapezoidal part, laminate material is being drawn in from two directions, leading to high intra-ply shear near that region, as illustrated in Figure 6 (bottom). Such high shear results in greater deviation in laminate thickness and fiber angles, which can potentially affect overall strength. There is a positive correlation between the amount and the direction of the drawn-in material with the mold geometry. As shown by Figure 6 (bottom), a large amount of material is drawn in from the sides and top of the laminate to form the side and sloped walls of the trapezoidal part. Laminate shear at the top and side walls was relatively minimal, but the surfaces transitioning from the side walls to the front sloped wall experienced the greatest shearing, which can adversely affect structural and surface quality. Based on the picture frame results from the preceding section, it is preferable for the laminate to undergo intra-ply shear below 45 degrees. Beyond this, much higher forces would be needed to shear the plies. Consequently, the laminate would shift to a less desirable bending mechanism for part forming. 

Figure 7 shows the stress contours and the laminate thickness distribution of the formed part. The alternating positive and negative normal and shear stresses near the edges of the sloped wall suggest potential wrinkling due to changes in curvature. Wrinkles may occur if the drawn-in material exceeds what is required to form the wall area. 

Figure 7 (bottom) also shows that these regions would naturally be thicker due to higher laminate shear and the potential for folds or wrinkles. Indeed, wrinkles are observable in our numerical simulation, particularly near the base of the sloped wall. Cuts or darts in the laminate (see dotted triangles in Figure 7 top right) can be made to facilitate material movement in preferred directions. This strategy helps mitigate stresses and reduces the likelihood of wrinkle formation, resulting in a stronger and longer-lasting part. 

Numerical simulation outputs such as traction magnitude, slip path length and contact pressure have demonstrated a positive correlation with observed defects on physically fabricated parts. Traction magnitude refers to the magnitude of in-plane traction, while slip path length indicates the relative slippage distance between the laminate and mold surfaces during the forming process. 

Figure 8 presents a comparison of traction magnitude and slip path length contour plots from numerical simulation alongside the physical defects observed in parts after thermoforming. The outer plies of the laminate experience sliding contact with tooling surfaces, resulting in tractions at these interfaces. Consequently, surface defects are often found in areas where significant tractions occur over a prolonged period. Wrinkles caused by draping constriction or friction lead to increased laminate thickness, resulting in traction lines, as depicted in Figure 8a. High tractions can also indicate optical distortions of the laminate surface and poor consolidation [18].

The slip path length (SPL) serves as a metric for gauging the duration of exposure of specific regions to the cooler tooling surface, thereby identifying areas prone to surface defects. Regions A and B in Figure 8b evidently demonstrate a positive correlation with matrix-smearing defects observed in the physically formed part. Contact slippage is expected as the molten laminate is draped and pressed by the molds. Such slippage should be minimized to avoid excessive localized shearing of laminate surfaces or even crazing (see Figure 8c) that may affect the overall part aesthetic and structural quality [14,19].

### Role of Laminate Bending and Shear Stiffnesses in Thermoforming Quality CFRTP Parts

A sensitivity study based on varying the original bending and shear stiffnesses of the laminate was numerically performed to investigate its effect on formability. The modeling setup is similar to that shown in Figure 1, and the 0-degree laminate orientation is used. The elastic young modulus and the constants eta_0_ and eta_inf_ in the shear-rate-dependent model were modified to 1/5th and five times their original value while keeping other laminate properties constant, and the response is as shown in Figure 9. 

Figure 10 shows the traction magnitude contours for the various cases. It was observed that with a constant laminate shear stiffness, a lower bending stiffness of the laminate would result in more wrinkling, particularly at the side and slope wall surfaces, as indicated in the top left figure. This is because it is relatively easier to initiate out-of-plane wrinkling than shear. However, this lower stiffness led to a reduced slip path length because the laminate was more compliant, allowing it to drape over the mold profile with less slippage and lower traction forces. In contrast, the stiffer laminate (with 5× bending stiffness) deformed more in shear, as shear deformation became the predominant mode. However, the side walls of the stiffer laminate experienced higher traction forces and greater slippage duration, which increased the likelihood of surface blemish.

The bottom row of Figure 10 illustrates how increasing shear stiffness affects the formability of the laminate while keeping the bending stiffness constant. Increasing shear stiffness has the same effect as reducing bending stiffness due to the competitive nature of the two deformation mechanisms. Consequently, when the bending resistance is diminished, the laminate becomes more prone to wrinkling, as shown by the bottom right figure. However, the relatively lower bending stiffness also enhances the laminate’s compliance, reducing traction forces and slippage between the laminate and the mold, which can potentially mitigate surface defects. The discussion suggests that competition between different deformation mechanisms exists, and their stiffness ratios could influence high-quality part production. Similar trends were observed for the 45-degree laminate orientation, albeit with less pronounced effects compared to the 0-degree case. Our conclusion is also aligned with that reported by Sebastiaan [3,4], who physically tested out laminate materials with various stiffness ratios. 

## 7. Thermoforming Modeling and Design Methodology 

This section details the general framework established for modeling and optimizing the thermoforming process of parts. Figure 11 illustrates the utilization of modeling to bolster the advancement of thermoforming processes across various stages of work. Examples are provided for each stage of the workflow, primarily aimed at elucidating the methodology and concept.

### 7.1. First Phase—Model Development and Ply Characterization

#### 7.1.1. Model Development

In this phase, the CAD for the mold of the part intended for thermoforming is created. This CAD geometry can be developed from any CAD software and should be checked for manufacturability. The selection of the part thickness is contingent upon the ply thickness, with consideration given to the intended strength or stiffness of the part. Typically, the part thickness is recommended to be an integer multiple of the ply thickness. For example, if the carbon fabric composite has a ply thickness of 0.3 mm, then the part thickness could be 0.6 mm or 1.2 mm. Based on the designated thickness, the male mold CAD can be offset by this value to create the female mold CAD with the required gap thickness. The CAD parts can be exported as STP, STEP or SAT files for meshing and then finally as STL or MSH format files to be imported into a thermoforming software such as AniForm^TM^.

Individual parts, such as the support frame, molds and the laminate, would have to be imported into AniForm^TM^ and assembled correctly before the spring tensioners from the support frame are added to the laminate. In setting up the model, the initial clearances between the laminate and the molds, the size and orientation of the laminate, the amount of pretension load and stiffness of the springs used, the hold time for the suspended laminate, and the location of the gripping spring tensioners on the laminate should be the same as intended in the physical set up. Trial simulations are then run to ensure analysis convergence and a reasonable range of results, including metrics such as traction, split-path length, shear angle, penetration depth and displacements.

#### 7.1.2. Ply Property Database Setup

Concurrently, to facilitate accurate thermoforming analysis, it is imperative to characterize the in-plane shear, bending, and ply–ply and tool–ply interfacial friction properties of the CFRTP plies. The properties can typically be obtained from the supplier, if available, or through independent testing procedures. While detailed testing procedures are not delineated here, as material testing is not the primary focus of this paper, such procedures are widely accessible in open literature sources [17,20,21,22]. 

Once these properties are obtained, the MatFit add-on in AniForm^TM^ is used to fit material models to the data that are acquired. The measurement data are pre-processed and stored in an ASCII-text file format compatible with MatFit. Once the measurement data are loaded, the MatFit tool allows for the selection of appropriate constitutive models that are fitted to the data via regression analysis. The best-fit material model and its parameters are saved as a material card. The characterization experiments are then simulated to ensure that the material models and their parameters accurately capture the response obtained from the characterization experiments. The preceding section on Materials and Methods (Figure 3) illustrates the validation of simulation results against experimental data for the picture-frame test, demonstrating close alignment of the shear force versus shear angle response between the modeled results and that obtained from experimentation.

### 7.2. Second Phase—Model Checks and Assisting in Thermoforming Design Stage

#### 7.2.1. Press Load and Laminate Sag Checks

For Phase 2 work, a series of checks are typically conducted by the finite element model to evaluate the feasibility of the physical thermoforming setup. One of these checks involves comparing the press load used during actual thermoforming with the press load obtained from simulation. As shown in Figure 12, the estimated press load from the simulation is around 11,000 N. Considering that the maximum capacity of the press machine is 65 ton-force, which is more than 52 times higher than that from simulation as a safety factor, the simulation aids in verifying whether the press machine is capable of thermoforming the part of a specific size and laminate thickness. 

It is evident that when the laminate is heated, its original stiffness diminishes, leading to sagging or out-of-plane deflection. This sagging effect is exacerbated by the weight of the laminate, stretching the spring tensioners and causing downward displacement, as depicted in Figure 13, which can result in uneven heating of the laminate.

As the spring tensioners are effectively 1D elements that provide stiffness only in the spring axial direction, they do not resist the out-of-plane displacement of the laminate. Consequently, if the displacement is significant, the bottom surface of the laminate may come into premature contact with the mold during transport to the forming stage prior to the forming process. Such premature localized contact and sliding can lead to uneven cooling and surface blemishes on the formed part. It is also worth noting that factors such as the stiffness of the tensioner springs, their locations, and their numbers, as well as the orientation of the laminate, can influence the degree of sagging. Hence, numerical modeling serves to predict the resulting clearance to avert premature contacts and assists in establishing guidelines for the required pretension and spring stiffness needed to maintain the laminate’s flatness after heating, thus mitigating the undesired contact issue. 

The correlation of the amount of laminate sagging between simulation and experiment can also help in further calibration of the ply bending properties. For instance, in this project, only the picture frame test was conducted to characterize ply shear properties, while default ply bending properties from a material card with a similar woven structure and matrix were utilized. The ply bending test was not conducted due to equipment and resources being unavailable. Instead, the laminate was suspended using different attachment configurations, such as 45 degrees and 90 degrees spring tensioner attachment, and for different hold times after preheating. The resulting laminate sag profile was then compared with simulation results to fine-tune the elastic and viscoelastic bending properties of the ply. This feedback or fine-tuning process is depicted as a dotted blue line from the Laminate Sag Check to Ply Properties Material Database in Figure 11. 

#### 7.2.2. Obstruction and Formability Checks

The obstruction check entails examining the forming simulation to identify any potential hindrances, such as attached spring tensioners, that may impede the complete closure of the mold. Figure 14 (left) shows that during forming, the laminate edges could be drawn within the footprint of the mold, causing the spring tensioners to be clamped by the mold. This suggests that either the laminate lacks sufficient excess material or that the mold surface is too large, resulting in obstruction that could potentially damage tools and equipment. Numerical simulations can promptly determine the necessary amount of excess material and cut-out laminate shape required to prevent such obstruction issues.

The virtual forming simulation enables the evaluation of whether a specific mold configuration or orientation relative to the laminate can completely form the part. Figure 14 (right) shows an attempt to form a part using a complex mold with two crevices. The mold is tilted roughly at 45 degrees to each of the formed crevices. Simulation results indicate that, despite the use of unobstructed clam-shell molds and restraining spring tensioners, the part will be under-formed. For such complex parts, potential solutions may involve forming simpler shapes and then assembling them or exploring alternative mold orientations for forming. 

#### 7.2.3. Mold Gap Check

Finally, the mold gap check is part of the mold design. Unlike parts made from purely polymers or composites with chopped fibers, which generally exhibit uniform thickness, fabric laminate composites display varying thickness distributions across the formed part, as depicted in Figure 15 (left). The thickness variation across the part could be as high as 50%, depending on the laminate orientation and gripping angles used. As a result, when the mold closes to form the part, some regions may experience higher contact pressure and traction than others. Regions with adequate contact pressure allow the molten matrix within the laminate to flow toward the part surface and make contact with the smooth mold surface to produce a glossy finish. Conversely, areas lacking sufficient contact pressure will yield a poorer surface finish. 

Figure 15 (right) illustrates the distribution of contact pressure on a part at the end of forming. Blue regions, typically at the curved edges, indicate a lack of contact between the laminate and the mold surface. Green regions, located at the tail of the gear knob, show low contact pressures, while orange and red regions indicate high contact pressures. This suggests that while the head of the gear knob may have a good surface finish, the tail may exhibit a poorer finish. These insights enable proactive adjustments to the mold design, such as slightly reducing the mold gap at the tail of the gear knob to ensure a more uniform contact pressure during forming, thereby enhancing the overall surface quality of the part.

Figure 16 (left) presents an example of a thermoformed gear knob cover characterized by a coarse, scaly surface quality resulting from inadequate contact pressure between the mold and laminate surfaces. To address this issue, the mold gap was decreased by 0.05 mm, enhancing confinement pressure and thereby producing a part with a shiny, glossy surface quality that is aesthetically pleasing, as shown in Figure 16 (right). While mold design is not the primary focus of this paper and is therefore not discussed in detail, it is worth noting that simulation can effectively support mold design improvements. 

### 7.3. Third Phase—Model Validation Stage

This phase of work involves verification of the modeled response against experimental measurement data. It includes comparing the laminate sag following infrared heating, the thickness distribution, the fiber orientation of the formed part and the correlation of indicators such as split-path length and traction magnitude with defect locations observed from experiments. Based on the quality of these correlations, fine-tuning of the material parameters within the simulation model may be conducted to enhance predictive reliability.

Figure 17 and Figure 18 illustrate the measurement locations on both the physical and virtual models of the trapezoidal and dome parts, which were thermoformed with the laminate oriented at 45 degrees to the mold axes. For each part, measurements were taken from three points on the front, side and top walls and used to compare the actual versus simulated results of the subtended angle between initially orthogonal fibers of the woven fabrics on the top layer of the laminate.

The ply shear angles from actual parts are approximated due to the curvatures and also matrix smearing on the surface, which makes exact determination of the ply angles difficult. The results of the ply shear angles for the 45-degree laminate cases are provided in Table 3. 

For the side wall of the dome profile, simulation shows points A and B have higher ply shear than point C, which is verified with measured data showing the same trend. For the front wall, the simulation predicts that point F near the base will experience the highest shear relative to points D and E, which is consistent with the trends observed from measured data. Additionally, the top wall of the formed dome part is shown to experience the least ply shear, as corroborated by both simulation results and measured data. 

For the side wall of the trapezoidal profile, both experimental data and simulation results indicate a higher shear at point A, near the slanted edge, and a lower shear at point C, which is furthest away from the front wall. On the top wall, both simulation and measurement show that ply shear angles are highest at point G, closest to the edge, and decrease progressively further away from the front wall.

Although the general trends align between simulation and experimental results, the magnitudes of the shear angles can vary considerably. These discrepancies are attributable to several factors, including variations in actual forming conditions compared to those modeled, temperature distribution across the laminate, highly nonlinear effects such as localized wrinkle formation and matrix smearing, and conditions at the actual mold/laminate interface. Specifically, the ply shear angles at the base of the front walls for both parts show poorer agreement because of the formation of wrinkles at that region. Such wrinkles reduce the observed ply shear angle significantly as the laminate undergoes folding instead of being sheared during the forming process. 

The thickness distribution of the formed polymer composite part was also compared between the experiment and simulation. A three-dimensional (3D) scanning of the CFRP part was performed using the Zeiss’s GOM ScanBox 6130 scanner in ARTC (Singapore), which captured measurements of both the top and bottom part surfaces, and the data were processed using Polyworks^TM^ (Version 2022) software to generate a 3D CAD model for cross-section thickness analysis. Figure 19 shows the GOM ScanBox equipment and some scanned CFRP samples.

Figure 20 presents a comparison between the thickness measurements taken from specific locations on the physical part and those derived from the simulation. It is crucial to recognize that the measurements from Polyworks™ software are based on the approximate stitching of the top and bottom surfaces. Furthermore, the results from the simulation indicate a nearly perfect symmetry in the thickness distribution, contrasting with the physical part, where less ideal forming conditions, such as slight variations in grip angle, pretension, or laminate orientation, can affect the final outcome. Consequently, comparing the qualitative trend, which exhibits the same order of magnitude, is more appropriate than a direct quantitative comparison, given the inherent limitations of the measurement technique. Nevertheless, thickness predictions at various locations on the formed parts were within 11.6% error from measurement. However, at two points with significant intra-ply shear, the prediction errors increased to around 20%. 

### 7.4. Fourth Phase—Parametric Study and Process Optimization

With the model validated against experimental data, this phase focuses on conducting a parametric study to investigate the effects of various processing parameters on the thermoforming performance of composite materials. It involves systematically altering factors such as laminate orientation, forming rate, temperature profiles and mold configurations to identify relationships and optimal conditions that enhance the quality and efficiency of the formed parts. For the study, the laminate is thermoformed using a female mold press, and the CFRTP laminate properties are provided in Section 2.

#### 7.4.1. Effect of Ply Orientation

Two forming analyses were conducted, one with the plies oriented at 0 degrees and another at 45 degrees relative to the mold axis, while maintaining fixed settings for preload, spring stiffness and ramp rate at 5 N, 0.1067 N/mm and 25 mm/s, respectively. Figure 21 presents the numerical results concerning slip path length, traction magnitude and thickness distribution on the formed part.

The analysis reveals that positioning the laminate at 45 degrees to the mold axis significantly reduces the likelihood of matrix smearing or wrinkling defects when forming the trapezoidal part. The 0-degree laminate orientation resulted in significant thickness deviations at the triangular surface adjacent to the slope wall, while the 45-degree orientation led to similar deviations at the bottom region of the sloped wall. The increased thickness is attributed to the development of intra-ply shear within these regions. Although the latter case displayed a wider spatial variation in laminate thickness, as evident on the sides and top of the formed part, it exhibited less absolute thickness increase compared to the 0-degree case. This suggests that adjusting the laminate angle can improve the structural consistency and quality of the component. As the laminate is suspended at the four corners (see Figure 1), the 45-degree laminate generates a shear zone at the center, allowing more intra-ply shear deviations during forming. In contrast, the 0-degree oriented laminate experiences limited intra-ply shear and results primarily in out-of-plane bending. This bending is less capable of material adjustment, particularly at part corners and concave recesses, making the laminate more prone to wrinkling and other defects.

#### 7.4.2. Effect of Ramp Rate

Two forming analyses were conducted using ramp rates of 25 mm/s and 80 mm/s, chosen to align with the operational limits of the thermoforming equipment at SIMTech. The preload, spring stiffness and laminate orientation were set at 5 N, 0.1067 N/mm and 0-degree, respectively, with the spring gripper locations illustrated in Figure 1. Figure 22 presents the numerical results for slip path length, traction magnitude and thickness distribution of the formed part.

The results indicated that for the specific laminate used, higher forming rates led to an increase in surface defects and wrinkles, as shown by an increase of around 15% and 10% of regions with higher slip path lengths and traction magnitudes on the contour plots, respectively. Generally, the forming speed must be fast enough to mitigate laminate cooling so as to press it into shape within the operating temperature window, but it must also be moderated to prevent the laminate from experiencing tearing. The optimal forming speeds are influenced by various factors, including the mold profile and the resin properties and weave structure of the carbon fabric. Our research has shown that shallower molds benefit from faster ramp rates, which contribute to producing a higher-quality product. In contrast, deeper molds, which involve more extensive shearing, stretching and bending of the laminate, coupled with potentially higher pressing loads, require a slower forming rate to ensure the production of a high-quality product. 

It was also observed that a faster forming rate results in a smaller increase in thickness at regions of high intra-ply shear, as illustrated in Figure 22. This phenomenon suggests that under fast-forming conditions, the laminate transitions away from intra-ply shear to other deformation mechanisms, largely attributable to the visco-plastic behavior of the molten laminate, which becomes more dominant in resisting shearing within the ply layers during the forming process. This apparently has a detrimental effect on the quality of the formed part, as shown by the contours in Figure 22, and whereby the intra-ply shear mechanism is generally preferred over bending for the thermoforming of laminated composites as indicated by references [3,4,5,23].

#### 7.4.3. Effect of Preload and Stiffness

Three forming analyses were conducted at different combinations of preloads and spring stiffness values, while other parameters, such as the laminate orientation and ramp rate, were kept constant. Figure 23 shows the numerical results of slip path length, traction magnitude and thickness distribution on the formed part. The results indicate that within the range of values tested, the impact of preload and spring stiffness on the forming process is negligible. It is generally understood that higher pretension would cause the laminate to stretch more before forming, which is required if the laminate sag is significant. However, this increased stretch can also lead to greater laminate distortion. Similarly, using a higher spring stiffness necessitates more force from the forming tools, which can lead to greater distortion. The main function of preload and tensioners is to suspend the laminate in position for forming; thus, excessive stress, which could harm the material, should be avoided. 

Conversely, if the pretension and spring stiffness are too low, the laminate may sag excessively, causing premature and prolonged contact between the molten laminate and the cooler mold, leading to the development of surface defects. The optimal settings for these parameters are also influenced by the drawn-in ratio of the part’s profile. It was found that shallower parts typically benefit from lower values of these parameters to minimize distortion, while deeper parts with more material that is drawn into the mold require higher values of pretension and spring stiffness to ensure better form quality. 

#### 7.4.4. Effect of Cuts and Darts

Cuts and darts (where the material is removed rather than simply cut) can be implemented on the laminate to reduce excessive material draw-in and mitigate issues such as wrinkling and surface defects. Typically, such modifications should be positioned close to the areas of concern, which, for a trapezoidal part, would be along the two sides of the sloped front wall. Figure 24 presents the numerical results of slip path length, traction magnitude, shear stress and thickness distribution for both the reference case and a case where two darts are placed at the top edge of the laminate. The results indicate that strategic use of darts on the laminate can enhance the quality of the part, as evidenced by the reduced intensity of the various metrics.

However, such cuts or darts can inadvertently cause increased laminate sagging as some regions no longer support the weight of the laminate. This can adversely affect the forming process. Additionally, the dotted lines on the top left picture show that if the laminate is cut at those specific regions to facilitate proper material draw-in, then the two top spring tensioners would need to be relocated, which might be constrained by the setup of the equipment, presenting further challenges to the forming process. 

## 8. Conclusions

A detailed modeling framework for the thermoforming of polymer matrix woven laminate composite was developed. Two key numerical indicators, slip path length and traction magnitude, were identified as positively correlated with matrix smearing and wrinkling defects. The material model was calibrated using picture-frame experimental results, and its prediction accuracy for intra-ply shear and thickness distribution was validated with measurements on physically formed parts. 

A parametric study was conducted to elucidate the relationship between various process parameters and the quality of the formed part. For the two profiles studied, sensitivity analysis revealed that the most to the least significant parameters influencing defects are laminate orientation, grip size, ramp rate, preload and tension stiffness. For the trapezoidal part, orienting the laminate at 45 degrees to the mold axis decreases the likelihood of matrix smear and wrinkling defects. This orientation, while causing greater spatial variation in part thickness, resulted in lower thickness deviation compared to the 0-degree orientation. Additionally, higher forming rates were observed to increase defects, attributed to greater slip path lengths and traction magnitudes. It is crucial that optimal forming speeds strike a balance between rapid shaping within the operational temperature window and avoiding material tear. 

Our study revealed that shallower molds benefit from faster ramp rates, whereas deeper molds require slower rates to accommodate extensive shearing, stretching and bending. Increased forming rates were also noted to result in smaller thickness increases at regions of high intra-ply shear, indicative of a transition from intra-ply shear to out-of-plane bending due to the visco-plastic behavior of the molten laminate. Finally, the strategic use of darts was demonstrated to enhance part quality by reducing the magnitude of defect indicators, highlighting the importance of thoughtful design adjustments in the thermoforming process.

## Figures and Tables

**Figure 1 polymers-16-02186-f001:**
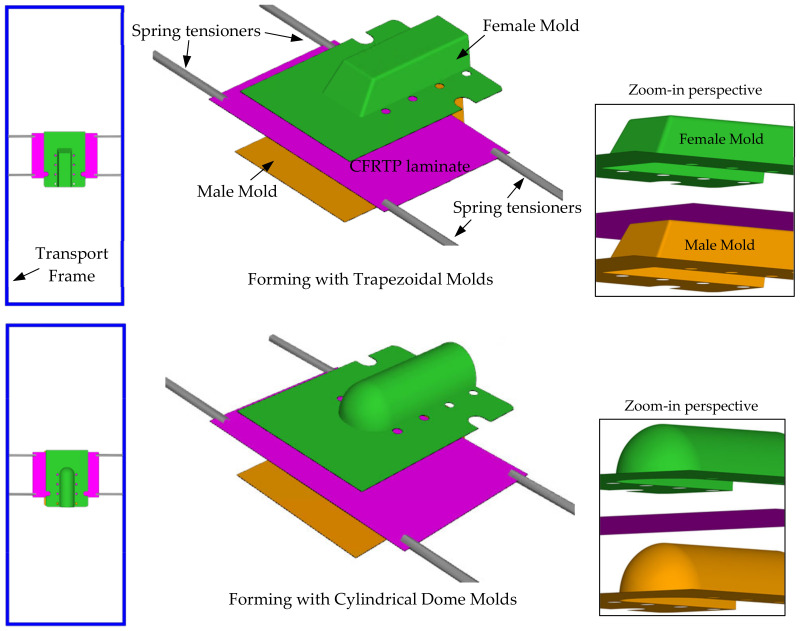
Overview of the thermoforming model setup.

**Figure 2 polymers-16-02186-f002:**
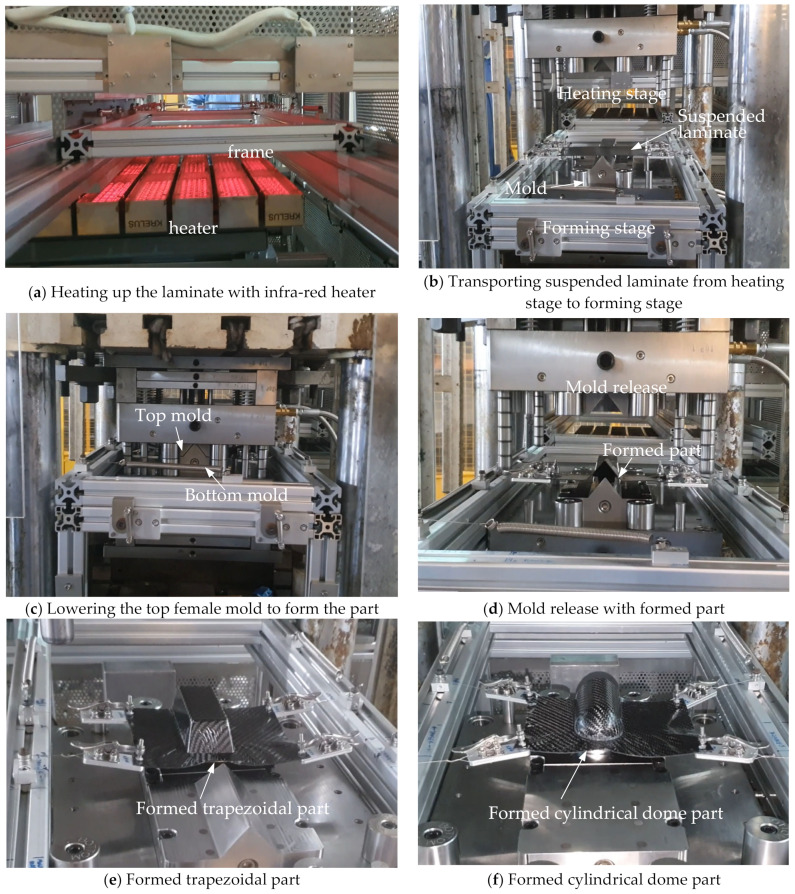
Physical thermoforming process as performed in SIMTech.

**Figure 3 polymers-16-02186-f003:**
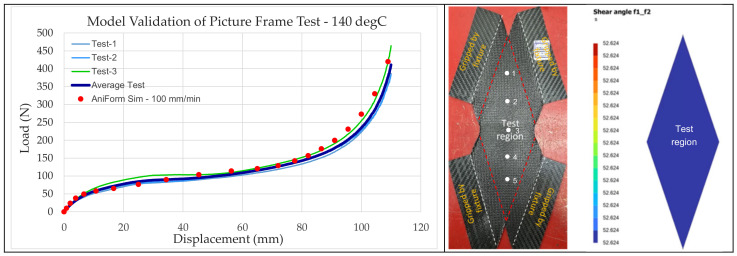
Close correlation of shear load–displacement curves between simulation and test.

**Figure 4 polymers-16-02186-f004:**
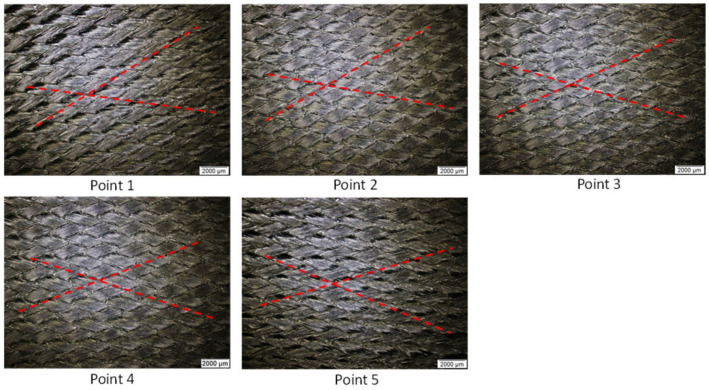
Measurement of ply shear angle from optical micrographs.

**Figure 5 polymers-16-02186-f005:**
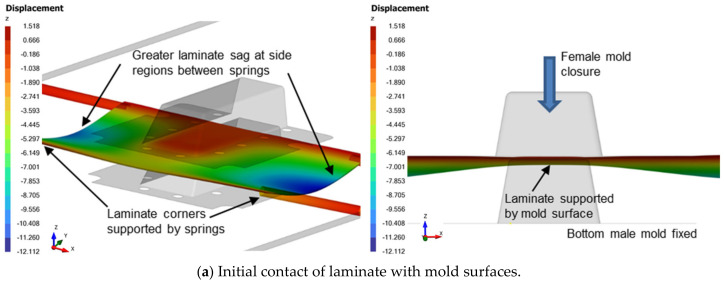
General laminate deformation process (left: isometric view, right: front view).

**Figure 6 polymers-16-02186-f006:**
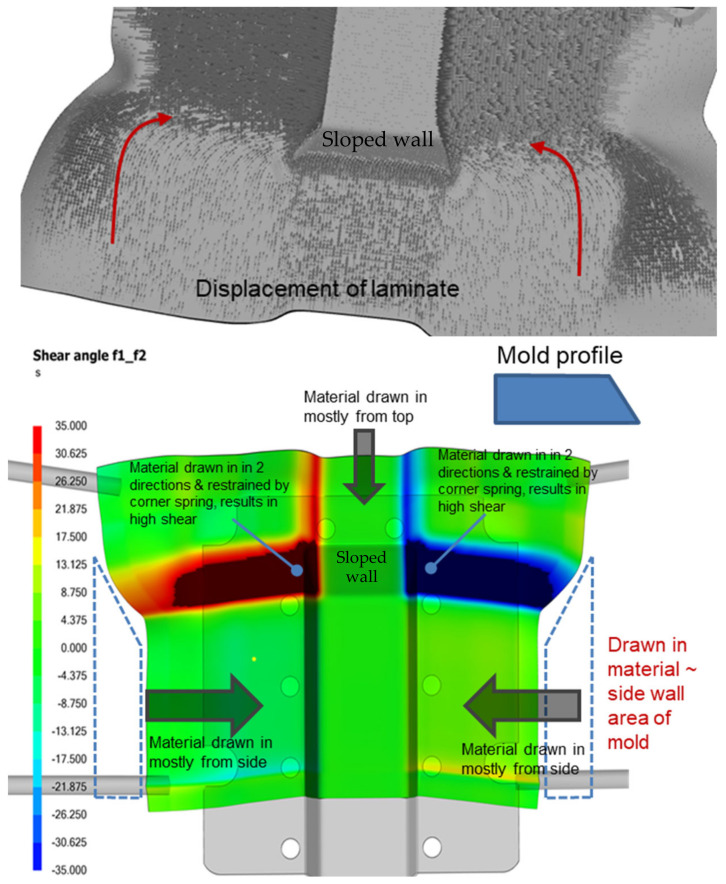
Example of thermoforming causing high distortion and laminate shear (0-deg laminate orientation).

**Figure 7 polymers-16-02186-f007:**
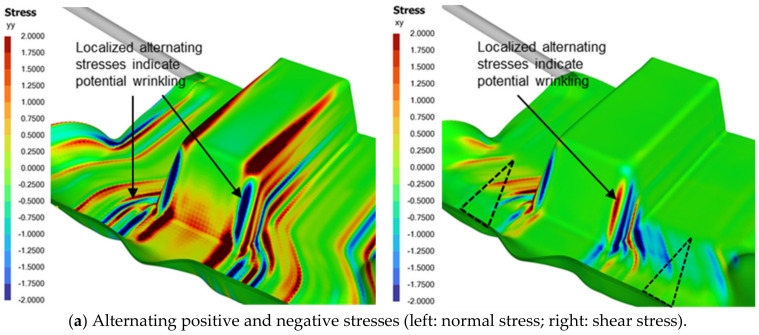
Stresses and thickness distribution on the formed part.

**Figure 8 polymers-16-02186-f008:**
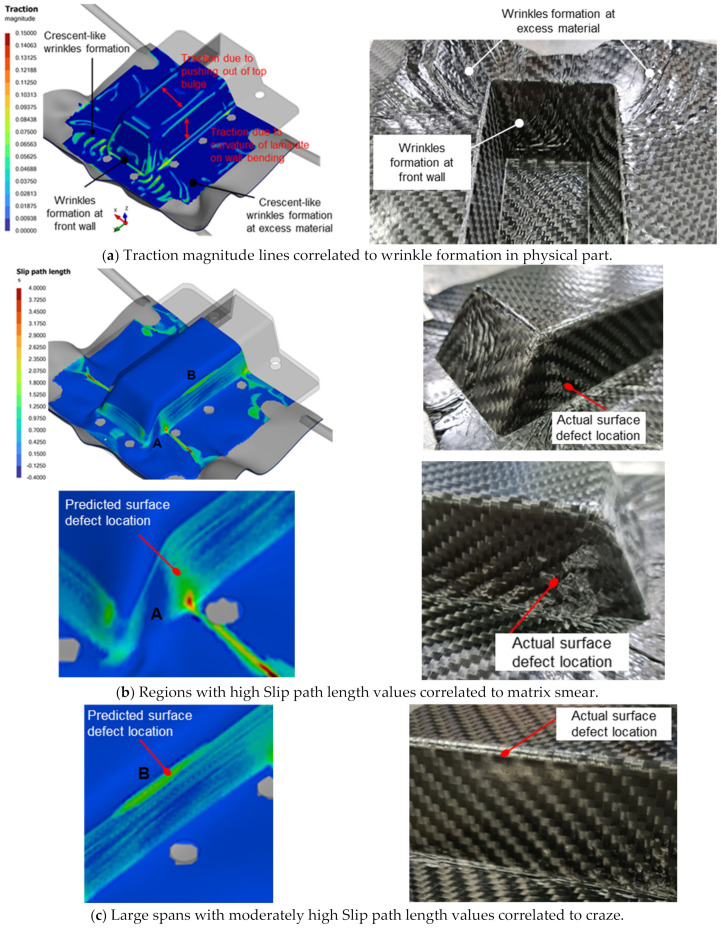
Positive correlation between simulation results and actual defects.

**Figure 9 polymers-16-02186-f009:**
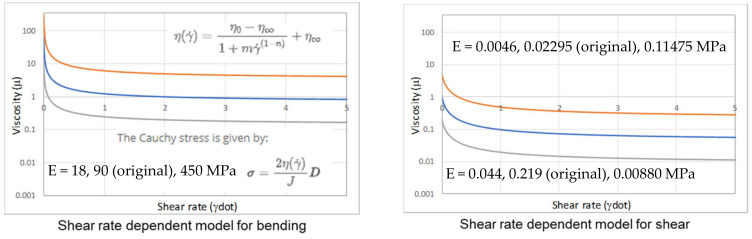
Laminate response of different bending stiffness (**left**) and shear stiffness (**right**). Blue: original values; grey: 1/5th of original values; orange: 5× original values.

**Figure 10 polymers-16-02186-f010:**
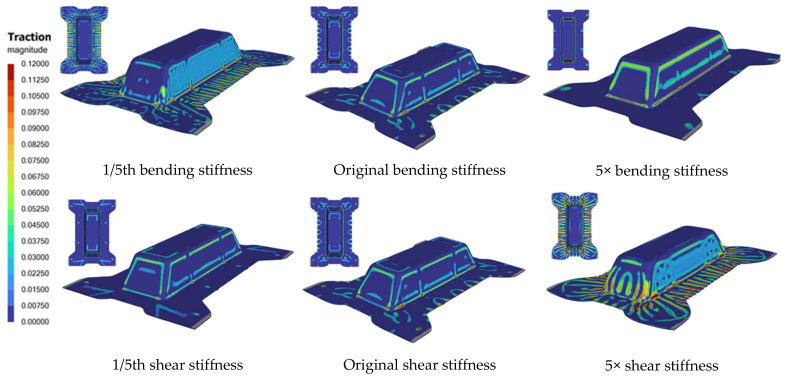
Effect of different bending (**top**) and shear (**bottom**) stiffnesses on traction magnitude contours.

**Figure 11 polymers-16-02186-f011:**
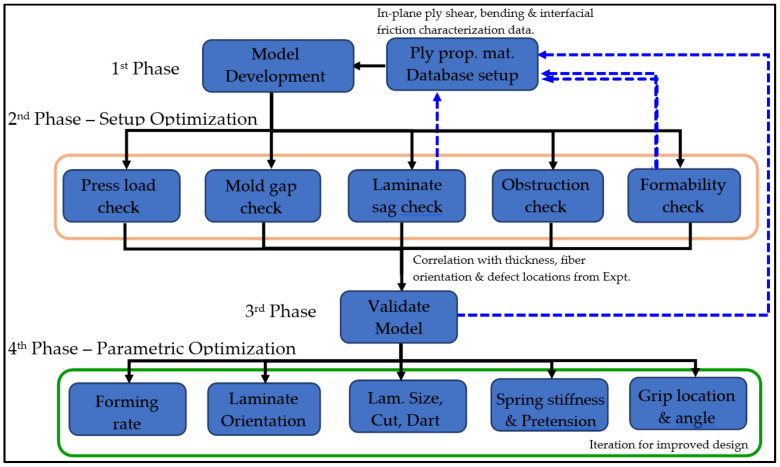
Modeling and process optimization workflow for composite thermoforming.

**Figure 12 polymers-16-02186-f012:**
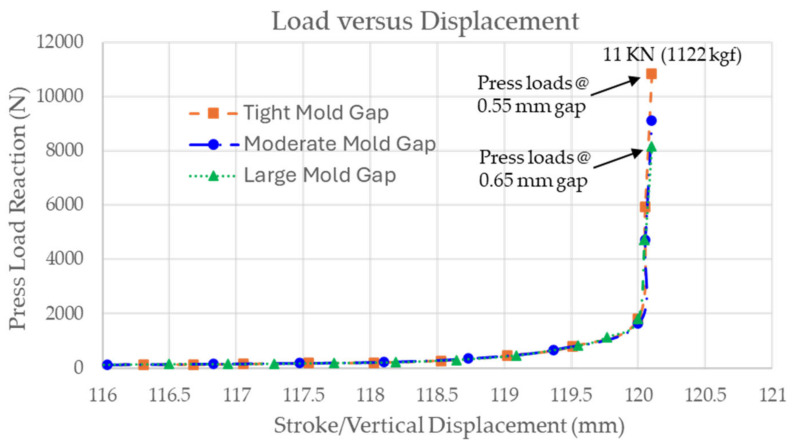
Estimated press load from simulation with different mold gaps (**top**) versus press load capacity of machine (**bottom**, dotted line box).

**Figure 13 polymers-16-02186-f013:**
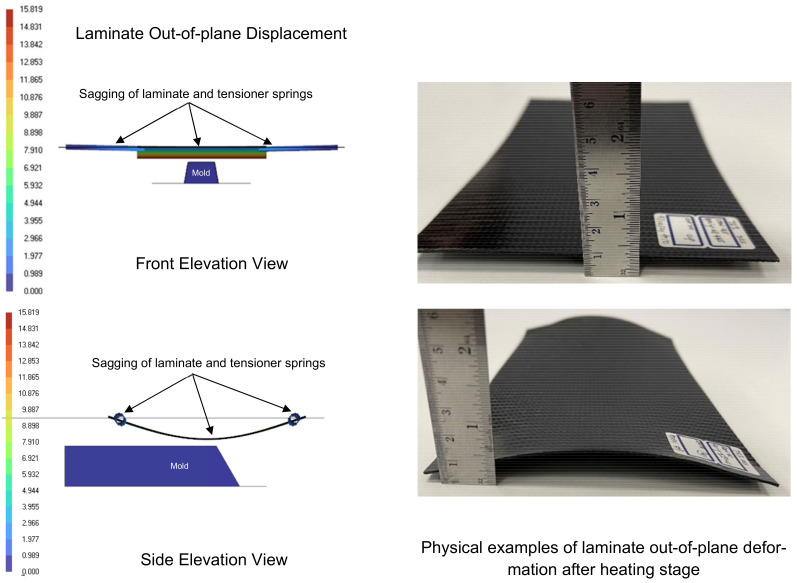
Laminate/tool premature contact check: estimation of mold clearance and spring tensioner requirements.

**Figure 14 polymers-16-02186-f014:**
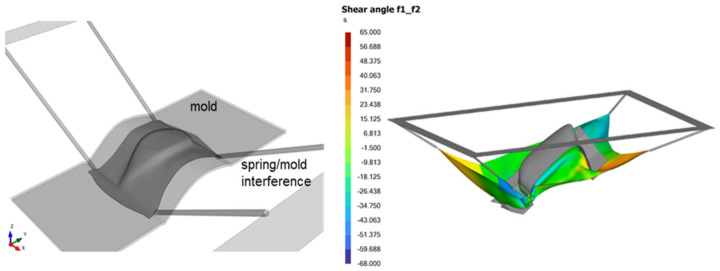
Example of virtual forming analysis used to detect obstruction issues and incorrect mold orientation resulting in unformable parts.

**Figure 15 polymers-16-02186-f015:**
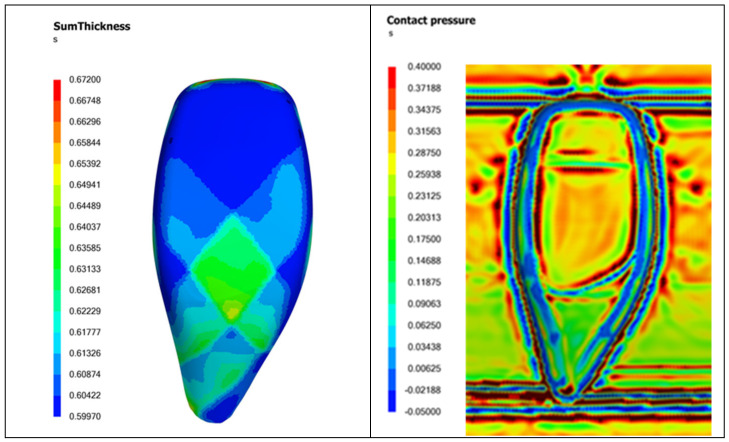
An example of part thickness (**left**) and contact pressure (**right**) variations with woven composite laminate thermoforming.

**Figure 16 polymers-16-02186-f016:**
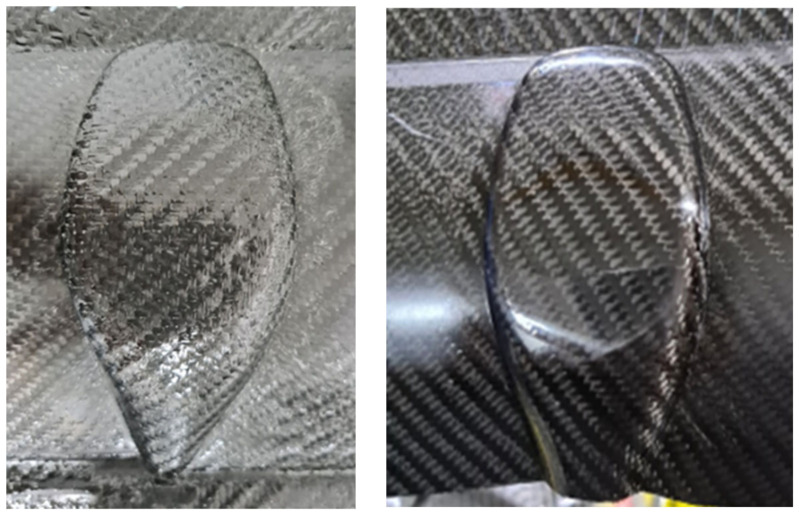
Comparison of scaly surface (**left**) and smooth glossy (**right**) finish of gear knob; an example of mold/part contact issue that can affect surface finish.

**Figure 17 polymers-16-02186-f017:**
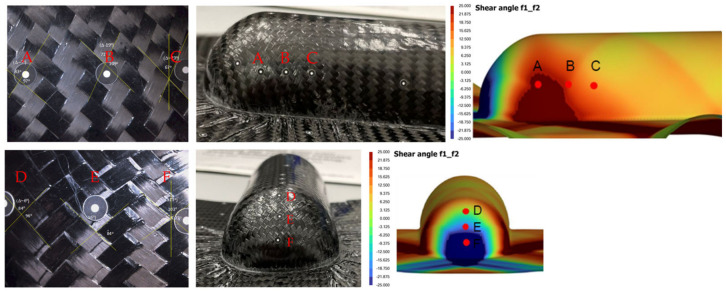
Measurement locations for ply shear angle on the dome part: (**top**) side wall; (**bottom**) front wall.

**Figure 18 polymers-16-02186-f018:**
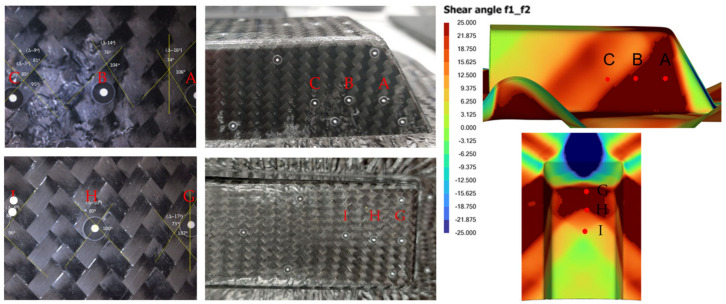
Measurement locations for ply shear angle on the trapezoidal part: (**top**) side wall; (**bottom**) top wall.

**Figure 19 polymers-16-02186-f019:**
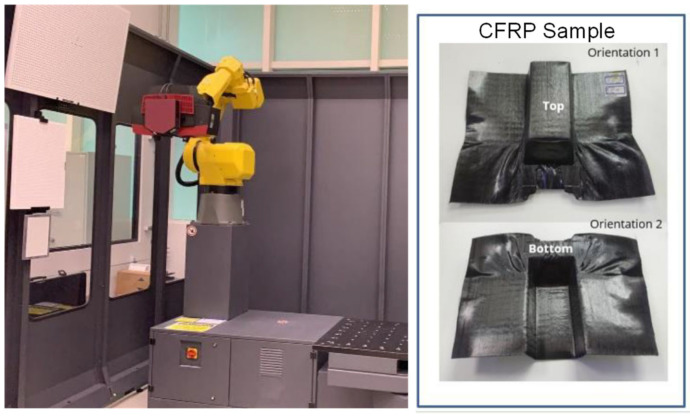
GOM ScanBox 6130 scanner (**left**) and scanned samples after thermoforming (**right**).

**Figure 20 polymers-16-02186-f020:**
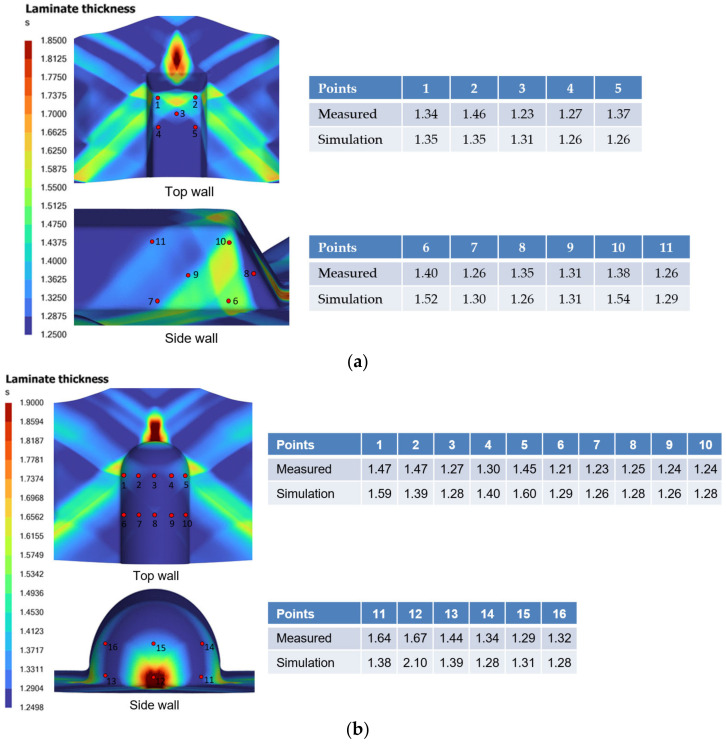
(**a**) Comparison of laminate thickness between measurement and simulation results at different locations of the trapezoidal part. (**b**) Comparison of laminate thickness between measurement and simulation results at different locations of the dome part.

**Figure 21 polymers-16-02186-f021:**
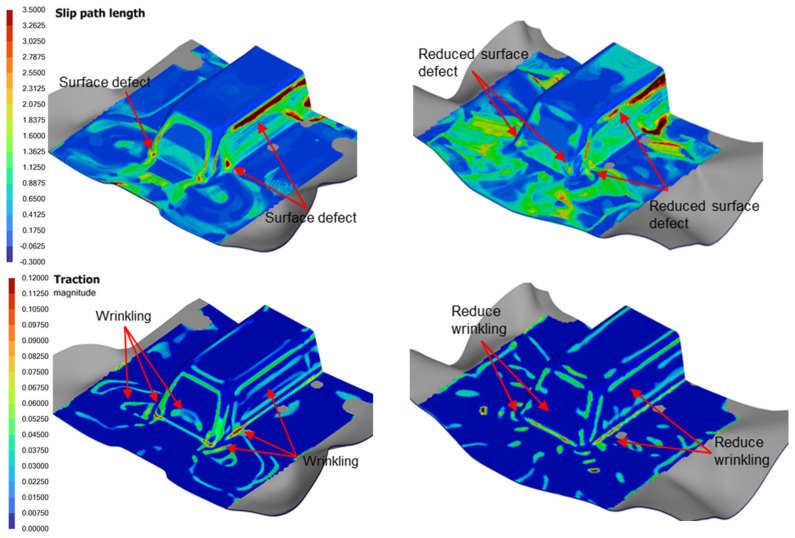
Comparison of defect indicators for different laminate orientation cases.

**Figure 22 polymers-16-02186-f022:**
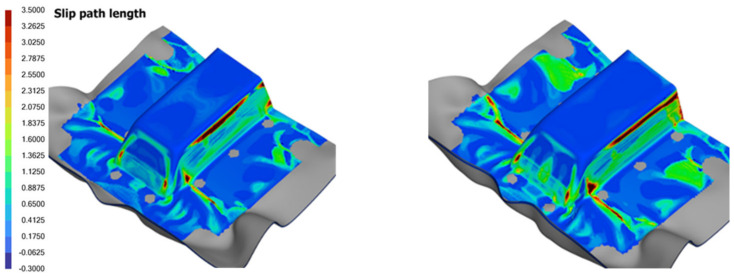
Comparison of defect indicators and laminate thickness for different forming rates.

**Figure 23 polymers-16-02186-f023:**
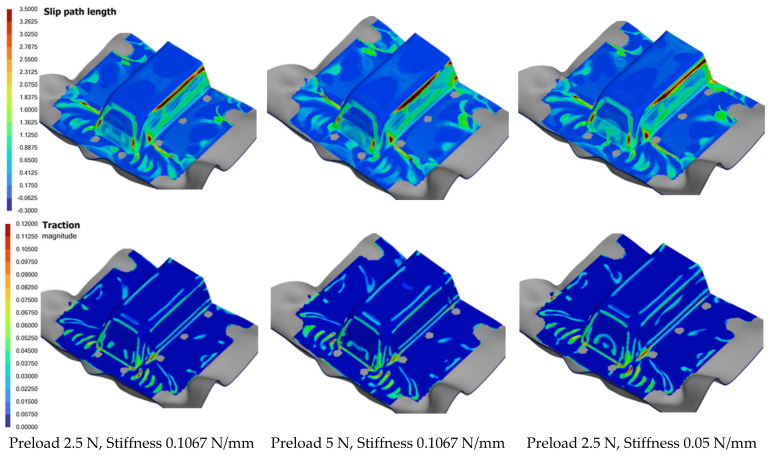
Comparison of defect indicators for different pretension and spring stiffnesses.

**Figure 24 polymers-16-02186-f024:**
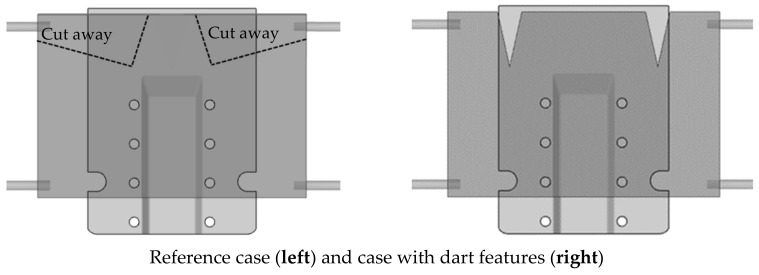
Comparison of defect indicators, shear stresses and thickness distribution for laminate with and without dart features.

**Table 1 polymers-16-02186-t001:** Material properties.

Property	Type	Parameter	Value	Unit
General Properties	Isotropic Elastic	υ	0	N/A
	Density	ρ	1780	kg/m^3^
In-plane Model	Fiber	E_F_	10000	MPa
	Isotropic Elastic	E_TI_	0.02295	MPa
	Cross-Viscosity	*η* _0_	1.0212	MPa.S
		*η_∞_*	0.043712	MPa.S
		M	17.0059	N/A
		N	0.074722	N/A
Bending Model	Isotropic Elastic	E_TB_	90	MPa
	Cross-Viscosity	*η* _0_	104.55	MPa.S
		*η_∞_*	0.683	MPa.S
		M	188.13	N/A
		N	0.1904	N/A

**Table 2 polymers-16-02186-t002:** Thickness and shear angle measurements.

**Sample 1**					
Location	Thickness			Ave. Thickness	Shear angel
	1	2	3		
1	0.605	0.628	0.594	0.609	49
2	0.595	0.592	0.616	0.601	50
3	0.639	0.650	0.641	0.643	48
4	0.591	0.616	0.601	0.603	53
5	0.601	0.593	0.608	0.601	51
**Sample 2**					
1	0.696	0.692	0.694	0.694	52
2	0.636	0.635	0.632	0.634	51
3	0.660	0.638	0.647	0.648	53
4	0.594	0.601	0.607	0.601	53
5	0.649	0.632	0.626	0.636	50
**Sample 3**					
1	0.621	0.634	0.645	0.634	50
2	0.604	0.592	0.610	0.602	50
3	0.638	0.619	0.621	0.626	50
4	0.606	0.612	0.632	0.617	51
5	0.638	0.621	0.648	0.635	51

**Table 3 polymers-16-02186-t003:** Comparison of outer-ply shear angle of formed parts between simulation results and measurement data for 45-degree laminate orientation and spring tension angle of 90 degrees.

**Sample**	**Lam. Angle**			**Spring tension angle**			**Feature**			Wrinkling on Frontwall 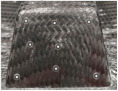
**A**	**45**			**90**			**Dome profile**			
**Ply Angle Data**										
**Wall Loc.**	Side (A)	Side (B)	Side (C)	Front (D)	Front (E)	Front (F)	Top (G)	Top (H)	Top (I)	
**Measured**	23	19	7	6	−9	−25	5	−3	0	
**Simulation**	25	22	12	6	−6	−13	2	0	0	
**Sample**	**Lam. Angle**			**Spring tension angle**			**Feature**			
**B**	**45**			**90**			**Trapezoidal profile**			
**Ply Angle Data**										
**Wall Loc.**	Side (A)	Side (B)	Side (C)	Front (D)	Front (E)	Front (F)	Top (G)	Top (H)	Top (I)	
**Measured**	23	14	7	12	9	−14	25	16	3	
**Simulation**	23	16	12	16	11	−20	20	14	6	

## Data Availability

The original contributions presented in the study are included in the article, further inquiries can be directed to the corresponding author.

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
