# Peer review of "A Modeling Framework for the Thermoforming of Carbon Fiber Reinforced Thermoplastic Composites"

_polymers, 2024, doi:10.3390/polym16152186_

Round 1

Reviewer 1 Report

Comments and Suggestions for Authors

An interesting research work has been carried out and some interesting results have been obtained. The following comments can be considered to improve the paper. 

1.       It is recommended to provide some quantitative results and findings in the abstract section. In addition, please provide on the mechanism of the effect of different process parameters on material properties.

2.       Introduction, it is suggested to provide the fabrication process, product type, mechanical performances, outstanding advantages (high toughness, high durability, fatigue resistance, repeated molding) as well as some potential application on fiber-reinforced thermoplastic composites compared to traditional thermosetting resin composites. In addition, the effects of related process parameters on the mechanical and long-term properties of thermoplastic composites should be further reviewed. The following are some relevant research work suggestions, which are further reviewed and supplemented, such as Construction and Building Materials, 2024, 429: 136455. Polymers, 2022, 14:2246.

3.       Part 5 presents materials and methods, and it is recommended that some of the results and discussions related to this part be included in the next part, such as figure 3.

4.       Some images of the simulation results in Part 6 are unclear, and it is recommended to replace them with high-definition images.

5.       How do you verify the accuracy of the simulation results in Part 6?

6.       As shown in FIG. 19, for an irregular shape, how to evaluate the mechanical properties of the material between the parts?

7.       After the optimization of the parameters, it is recommended to provide a simple summary to analyze which parameters are more helpful for obtaining better material properties.

Author Response

Please see the word document attachment.

Thank you so much. 

Reviewer 2 Report

Comments and Suggestions for Authors

Authors presented "A Modeling Framework for the Thermoforming of Carbon Fiber Reinforced Thermoplastic Composites", which sounds interesting.

The presentation of the data is superb. However, it's unlikely that Polymers has the proper audience for this research. One of the engineering journals has to get this article.

English needs a little revision.

Formating errors:

1. Spelling mistake in Figure 2f caption.

2. Line # 186 spelling mistake (should be molding in place of modelling). 

3. Word "A." is written on Figure 5b, explanation required.

4. The font style in Figures 3 and 11, as well as Graph 12, ought to be modified to align with the other font type used in the study.

Comments on the Quality of English Language

A slight revision required

Author Response

(The authors gave the same response as above.)

Reviewer 3 Report

Comments and Suggestions for Authors

Authors develop thre comprehensive modeling framework to  thermoform of polymer matrix woven laminate composite. The main idea of paper is to suggest two numerical indicators, namely the slip path length and traction magnitude, which can positively correlated to matrix smearing and wrinkling defects. The model was calibrated with experimental results, and the prediction accuracy for intra-ply shear and thickness distribution were examined. Additionaly, authors perform a parametric investigation to  obtain the relationship between various process parameters and the quality of the formed parts. Paper can be accepted in the present state.

Author Response

(The authors gave the same response as above.)

Reviewer 4 Report

Comments and Suggestions for Authors

The authors present an innovative study that can have great value for enginerss and practitioner. I have no doubt on the value of the article and I only propose some minimal changes:

Figure 1,

Perhaps increasing the distance of the elements in the isometric view can increase the figure's value. I propose a lower scale representation at the right of the actual figures.

Add information about the thickness of the molds and the used draft angles.

Table 1, define the acronyms, E, M…

Table 2 is difficult to read. I lack information on the position of the points and the significate of 1, 2, 3 ( I understand that are three measures made at the same point, but it is my intuition)

Figure 5, Eliminate “A” from inside 5b figure.

Some of the defects detected in the trapezoidal model are typical of a design poorly suited for the manufacturing process. Large rounding favors the drawn of the material. This is known for thermoplastic thermoformed and blow-molded parts.

In some figures the scale for the FEM is difficult to read due to pixelation. Example figure 22

Author Response

(The authors gave the same response as above.)

Round 2

Reviewer 1 Report

Comments and Suggestions for Authors

Accept!